# Circadian Rhythm and Sleep Analyses in a Fruit Fly Model of Fragile X Syndrome Using a Video-Based Automated Behavioral Research System

**DOI:** 10.3390/ijms25147949

**Published:** 2024-07-20

**Authors:** Sara Milojevic, Arijit Ghosh, Vedrana Makevic, Maja Stojkovic, Maria Capovilla, Tomislav Tosti, Dejan Budimirovic, Dragana Protic

**Affiliations:** 1Department of Pharmacology, Clinical Pharmacology and Toxicology, Faculty of Medicine, University of Belgrade, 11000 Belgrade, Serbia; sara.milojevic@med.bg.ac.rs (S.M.); maja.stojkovic@med.bg.ac.rs (M.S.); 2Chronobiology and Behavioral Neurogenetics Laboratory, Neuroscience Unit, Jawaharlal Nehru Center for Advanced Scientific Research, Jakkur, Bangalore 560064, India; arijitghosh2009@gmail.com; 3Department of Pathophysiology, Faculty of Medicine, University of Belgrade, 11000 Belgrade, Serbia; vedrana.parlic@med.bg.ac.rs; 4UMR7275 CNRS-INSERM-UniCA, Institute of Cellular and Molecular Pharmacology Institute, Sophia Antipolis, 06560 Valbonne, France; maria.capovilla@ipmc.cnrs.fr; 5Institute of Chemistry, Technology and Metallurgy, National Institute of the Republic of Serbia, University of Belgrade, 11000 Belgrade, Serbia; tosti@chem.bg.ac.rs; 6Department of Psychiatry, Fragile X Clinic, Kennedy Krieger Institute, Baltimore, MD 21205, USA; budimirovic@kennedykrieger.org; 7Department of Psychiatry & Behavioral Sciences-Child Psychiatry, School of Medicine, Johns Hopkins University, Baltimore, MD 21205, USA; 8Fragile X Clinic, Special Hospital for Cerebral Palsy and Developmental Neurology, 11000 Belgrade, Serbia

**Keywords:** *Drosophila melanogaster* model of fragile X syndrome, *FMR1* gene, Fragile X syndrome, circadian rhythm, sleep

## Abstract

Fragile X syndrome (FXS) is caused by the full mutation in the *FMR1* gene on the Xq27.3 chromosome region. It is the most common monogenic cause of autism spectrum disorder (ASD) and inherited intellectual disability (ID). Besides ASD and ID and other symptoms, individuals with FXS may exhibit sleep problems and impairment of circadian rhythm (CR). The *Drosophila melanogaster* models of FXS, such as *dFMR1^B55^*, represent excellent models for research in the FXS field. During this study, sleep patterns and CR in *dFMR1^B55^* mutants were analyzed, using a new platform based on continuous high-resolution videography integrated with a highly-customized version of an open-source software. This methodology provides more sensitive results, which could be crucial for all further research in this model of fruit flies. The study revealed that *dFMR1^B55^* male mutants sleep more and can be considered weak rhythmic flies rather than totally arrhythmic and present a good alternative animal model of genetic disorder, which includes impairment of CR and sleep behavior. The combination of affordable videography and software used in the current study is a significant improvement over previous methods and will enable broader adaptation of such high-resolution behavior monitoring methods.

## 1. Introduction

Fragile X syndrome (FXS) is a neurodevelopmental disorder caused by an expansion of the CGG trinucleotide repeats (more than 200, termed full mutation, FM) in the 5′ untranslated region of the Fragile X Messenger Ribonucleoprotein 1 (*FMR1*) gene on the Xq27.3 chromosome region [1,2]. The *FMR1* gene codes FMR1 protein (FMRP), which is deficient in FM, and clinically manifests as FXS. FMRP is an RNA-binding protein important for the translation of certain mRNAs involved primarily in brain development, neuronal synapse formation, and synaptic plasticity [3]. FMRP is known as a translation repressor which limits the synthesis of various proteins in the brain. *FMR1* transcription is suppressed by hypermethylation of the *FMR1* gene, occurring in the FM. In the absence of FMRP, neuronal circuit formation and higher cognitive functions can be impaired [4]. FMRP plays a crucial role in synaptic function due to the regulation of the translation of the metabotropic Glutamate Receptor 5 (mGluR5) in astrocytes and to myelin production in oligodendrocytes [5,6,7]. On the other hand, FMRP can regulate transcription and RNA synthesis by interaction with transcription factors and chromatin-modifying enzymes [4]. Finally, FMRP can interact with some proteins such as ion channels, and regulate their function [4]. FMRP, through the interaction with the large conductance Ca^2+^-activated K^+^ (BK) channels, modulates action potential duration and consequently regulates neurotransmitter release and short-term plasticity in CA3 hippocampal pyramidal neurons [3]. All these factors together contribute to the clinical presentation of FXS.

The prevalence of FXS, in the general population, depends on geographical areas and assessed population. In general, prevalence is 1 in 8000 females and 1 in 5000 males [8]. FXS is the most common monogenic cause of autism spectrum disorder (ASD) and inherited intellectual disability (ID) [9]. IQ levels in patients with FXS can be correlated with gender, methylation status, and FMRP abundance [10,11]. Approximately 85% of males and 25–30% of females with FXS have ID [6]. On the other side, more than half (50–60%) of males and 20% of females with FXS are diagnosed with ASD [12] (reviewed in Protic et al., 2022 [13]). Generally, females with FXS have less severe symptomatology due to compensatory activation of the unaffected X [13,14].

Besides ASD, patients with FXS exhibit ID, sleep problems, changes in circadian rhythm (CR), delay in the development of motor functions, speech, and language in early childhood, and often attention deficit hyperactivity disorder (ADHD), anxiety, and aggression [15,16]. FXS can also be characterized by other comorbidities such as neurological seizures, endocrinological problems like obesity, cardiac abnormalities (mitral valve prolapse), and macroorchidism in puberty [17,18]. Recurrent otitis media infections in childhood and strabismus are also present in patients with FXS. Physical appearance can also be characterized by an elongated face, prominent ears, mandibular prognathism, and joint hypermobility [17]. As mentioned above, sleep problems are often present in individuals with FXS and manifest mostly as hardly falling asleep, impaired sleep quality, and waking up many times during the night. Sleep problems tend to affect the quality of a patient’s life and have a negative impact on the whole family [13,19].

FXS preclinical research on animals is limited to a few models, such as the *Fmr1* knock-out (KO) mice, the *Fmr1* KO zebrafish, and the *Drosophila melanogaster* model of FXS (*Fmr1,* FBgn0028734 [20]; herein named *dFMR1*) such as *dFMR1^B55^* mutants [21,22,23]. In *dFMR1^B55^*, 2.5 kb deletion including exons 2, 3, and 4 of the *dFMR1* gene was generated by an imprecise excision of the EP(3)3422 P-element [24]. The *Drosophila melanogaster* models of FXS, such as *dFMR1^B55^*, show impairment in CR, climbing abilities, social interactions, olfactory learning, and memory [25,26]. Tracking of locomotor activity enables sleep and CR analysis in the *Drosophila melanogaster* models of FXS. Until now, CRs have been studied in *Drosophila* mostly with infrared-based beam-crossing methods [24,26,27,28,29,30]. These methods may lead to an underestimation of the total activity of flies throughout the day, which may, in turn, affect estimation of critical CR and sleep statistics [31,32].

The aim of this study is to analyze sleep patterns and CR in *dFMR1^B55^* mutants as an animal model for further research in the field of FXS, using a new platform based on continuous high-resolution video monitoring in integration with a customized version of the open-source app VANESSA (available at https://zenodo.org/records/5558502, accessed on 8 July 2024).

## 2. Results

### 2.1. CR Analysis in the w^1118^, per^01^ and dFMR1^B55^ Groups

The CR of *white^1118^* (*w^1118^*), *period01* (*per^01^*), and *dFMR1^B55^* flies in the dark–dark (DD) period (from 3rd to 10th experimental day) were analyzed using video monitoring and VANESSA application.

A graphical representation of the activity patterns (CRs) of selected flies of the *w^1118^*, *per^01,^
*and *dFMR1^B55^* groups in the DD period is presented in actograms within Figure 1.

In the control *w^1118^* groups (N = 50 males and N = 50 females), rhythmicity was observed in 84% (42/50) of females and in 96% (48/50) of male flies. Comparison between sexes showed a significantly higher frequency of rhythmic flies in males (*p* = 0.04). The mean period for rhythmic individuals was 23.70 ± 0.17 h for females (N = 42) and 23.78 ± 0.27 h for males (N = 48) and there was no statistically significant difference in period values (*p* = 0.14) between sexes. In addition, *w^1118^* rhythmic females had significantly higher (*p* < 0.0001, both) median and mean values of rhythm power (median = 263.90, with range: 68.85–423.60; mean ± SD: 252.60 ± 100.90) in comparison with *w^1118^
*rhythmic males (median = 110.90, with range: 21.17–385.00, mean ± SD: 125.60 ± 72.49).

In the *per^01^* groups, rhythmicity was observed in 62% (31/50) of females and in 62% (31/50) of male flies. Comparison between sexes showed the same frequency of rhythmic flies (*p* > 0.99). The mean period for rhythmic individuals was 23.60 ± 0.31 h for females (N = 31) and 23.86 ± 0.27 h for males (N = 31). Furthermore, for rhythmic females, the median value of the power of the rhythm was 145.00 (range: 17.99–395.50) and the mean value was 157.30 ± 102.50. In addition, the median and mean values for rhythmic males were 68.49 (range: 15.10–373.80 and 109.60 ± 102.70, respectively. Interestingly, male *per^01^
*rhythmic individuals had a significantly longer period (*p* = 0.0002), while *per^01^
*rhythmic females had a significantly higher power of the rhythm than the rhythmic *per^01^
*males (*p* = 0.0180).

Finally, in the experimental *dFMR1^B55^* groups, rhythmicity was observed in 98% (49/50) of females and in 74% (37/50) of male flies indicating a significantly lower frequency of rhythmic flies in males (*p* = 0.0005). The mean period for rhythmic individuals was 23.66 ± 0.36 h for females (N = 49) and 23.69 ± 0.49 h for males (N = 37) without statistically significant difference in period values (*p* = 0.51) between sexes. The median values of the power of the rhythm were 128.90 (range: 14.18–314.80) for rhythmic females and 82.65 (range: 15.12–292.20) for rhythmic males, while the mean values of the same parameter were 135.60 ± 68.81 for females and 90.18 ± 65.16 for males. Like *w^118^* and *per^01^
*females, *dFMR1^B55^* rhythmic females had a significantly higher power of rhythm than the rhythmic *dFMR1^B55^* males (*p* = 0.001).

The described values of periods and power of the CRs, in individual rhythmic and arhythmic *w^1118^*, *per^01^*, and *dFMR1^B55^* males, are visualized within selected chi-square periodograms in Figure 2.

Comparisons of examined variables related to CR (frequency of rhythmicity, period, and power of rhythm) among included groups of flies revealed that the lowest frequencies of rhythmic flies of both sexes were noticed in *per^01^
*(61% in both, males and females). However, a significantly lower frequency of rhythmicity was shown for *dFMR1^B55^
*males compared to *w^1118^* males (*p* = 0.002). Interestingly, *dFMR1^B55^
*females had a higher frequency of rhythmicity compared to *w^1118^* of the same sex (*p* = 0.01). On the other hand, there were no statistically significant differences between *dFMR1^B55^* and *per^01^
*males (*p* = 0.20) in terms of frequency of rhythmic flies, while for females, *dFMR1^B55^* showed a higher frequency of rhythmicity compared to *per^01^ (p* < 0.0001*).* Finally, for both sexes, *w^1118^
*had a higher frequency of rhythmicity compared to *per^01^* (females: *p* = 0.01; males: *p* < 0.0001).

Periods for rhythmic individuals were not statistically different for *w^1118^* vs. *per^01^
*vs. *dFMR1^B55^* (females: *p* = 0.25; males: *p* = 0.10). Multiple comparisons among included groups divided by sex did not show statistically significant differences in period values (*p* > 0.05, all). Violin plots of periods, presented in Figure 3, show that there were approximately the same period values in rhythmic males of all examined groups *(w^1118^*, *per^01,^
*and *dFMR1^B55^*).

The power of the rhythm comparison among *w^1118^*, *per^01,^
*and *dFMR1^B55^* revealed statistically significant differences among them (females: *p* < 0.0001; males: *p* = 0.02). Specifically, for both sexes, *dFMR1^B55^* had significantly lower power than *w^1118^* (females: *p* < 0.0001; males: *p* = 0.03). There were no significant differences between *dFMR1^B55^* and *per^01^
*in power values (both sexes: *p* > 0.9999).

Bar plots of frequency of rhythmicity and power of rhythm in *w^1118^*, *per^01^*, and *dFMR1^B55^* are presented in Figure 4.

### 2.2. Sleep Analysis in the w^1118^, per^01^, and dFMR1^B55^ Groups

The sleep patterns of *w^1118^*, *per^01^*, and *dFMR1^B55^* flies were analyzed using video monitoring and the VANESSA application. Sleep was analyzed for the light–dark (LD) period (1st and 2nd experimental day), and for DD periods (3rd to 10th experimental day). Specifically, for the LD period, sleep was separately analyzed for daytime (DT; ‘lights-on’ period of the day; ZT00–ZT12) and nighttime (NT; ‘lights-off’ period of the day; ZT12 to ZT24).

In the control *w^1118^* groups, the total sleep time (TST) per day was 119.25 ± 120.17 min (median: 81 min, with range: 0–647 min) during DT and 409.81 ± 127.62 min (median: 416 min, range: 34–618 min) during NT for females. In addition, TST was 318.26 ± 155.96 min (median: 325 min, with range: 0–617 min) in DT, and 348.66 ± 162.56 min (median: 347 min; range: 5–638 min) in NT for males. Females had 10.18 ± 5.92 sleeping episodes/bouts (median: 10, with a range 1–29) in the DT and 21.53 ± 6.91 (median: 22, range: 6–37) in NT, while males had 16.11 ± 5.63 (median: 16, with a range 1–29) in DT and 23.30 ± 8.76 (median: 23.5; range: 1–44) in NT.

In the *per^01^
*groups, the TST per day was 38.67 ± 70.64 min (median: 8.5 min, with range: 0–370 min) during DT and 332.60 ± 146.64 min (median: 324 min, range: 18–630 min) during NT for females. For males, TST was 138.73 ± 150.49 min (median: 100 min, with range: 0–509 min) in DT and 405.44 ± 136.74 min (median: 418 min; range: 97–699 min) in NT. Females had 7.21 ± 7.55 sleeping bouts (median: 5, with a range of 1–32) in the DT and 10.18 ± 5.18 (median: 9.5, range: 2–30) in NT, while males had 12.79 ± 8.34 (median: 11, with a range 1–34) in DT and 14.34 ± 4.95 (median: 15; range: 5–32) in NT.

Finally, in the experimental *dFMR1^B55^* groups, the TST per day was 270.44 ± 133.37 min (median: 300 min, with range: 0–499 min) in DT and 447.26 ± 102.53 min (median: 458.50 min, range: 57–622 min) during NT for females. For males, TST was 357.65 ± 169.94 min (median: 414.50 min, with range: 13–616 min) in DT and 520.98 ± 81.72 min (median: 537 min; range: 231–672 min) in NT. Females had 16.15 ± 6.83 sleeping bouts (median: 16, with a range 1–31) in DT and 17.1 ± 8.41 (median: 16, range: 2–43) in NT, while males had 20.50 ± 8.68 sleeping bouts (median: 22, with a range 2–38) in DT and 15.96 ± 10.14 (median: 14, range: 1–47) in NT.

Additional sleep parameters that could provide a more comprehensive understanding of sleep disturbances in the FXS model such as sleep bout length, sleep latency, wake after sleep onset (WASO), and sleep fragmentation index (SFI), separately for females and males, in LD periods, are presented in Table 1 (for DT) and in Table 2 (for NT). These tables give a more holistic view of the sleep patterns in the different experimental groups of flies.

As presented in Table 1 and Table 2, a comparison of examined variables related to sleep revealed statistically significant differences among the three groups, in both males and females (*p* < 0.0001, all). The longest TSTs during DT and NT of LD periods were noticed in *dFMR1^B55^* males. In addition, *dFMR1^B55^* males and females had statistically significantly more frequent sleep episodes in DT in comparison with flies in the *w^1118^* and *per^01^* groups (*p* < 0.0001, all). Male flies in the *dFMR1^B55^* group had statistically significant shorter sleep latency in comparison with males from the *w^1118^* and per^01^ groups (*p* < 0.0001, all) in DT of LD period. The WASO was also statistically significantly shorter in *dFMR1^B55^
*males than the same parameter in males of the *w^1118^* and per^01^ groups (*p* < 0.0001, all) during NT of the LD period. Furthermore, the Mann–Whitney test revealed that there were statistically significant differences between male and female flies in the *dFMR1^B55^* group in DT of the LD period in terms of investigated sleep parameters values (males vs. females: number of bouts: *p* = 0.0002; sleep latency: *p* < 0.0001 and TST: *p* < 0.0001). Similarly, there were statistically significant differences between males and females in the *dFMR1^B55^* group in NT of the LD period (males vs. females: *p* = 0.0074 for bout length; *p* = 0.0032 for sleep latency; *p* < 0.0001 for TST and WASO, and *p* = 0.0075 for SFI). Finally, as presented in Table 1 and Table 2**,** analyzed sleep parameters show that *dFMR1^B55^
*flies spent statistically significantly more time sleeping per day. It is clear that both DT and NT sleep had been increased in *dFMR1^B55^
*mutants. Finally, the graphical presentation of average sleep profiles for the LD period (1st to 2nd day) of sleep analyses in *w^1118^*, *per^01^*, and *dFMR1^B55^
*groups are presented in Figure 5.

Additional sleep analyses were also performed for the DD period. The obtained results are presented in Appendix A. Please note that in the DD period (3rd to 10th experimental day) ‘lights-on’ time (marked as daytime-DT in the LD period) does not exist. Accordingly, the results are presented per day for the whole DD period without separate presentations for the first (DT: ZT00-ZT12) and the second (NT: ZT12 to ZT24) parts of the day.

## 3. Discussion

The current study innovatively used high-resolution continuous video monitoring for a detailed analysis of circadian rhythm and sleep patterns in *Drosophila Fmr1* mutants based on their locomotor activity. This study revealed a lower percentage of rhythmic males in the *dFMR1^B55^* group in comparison to the control wild-type male flies. This observation may be based on the duration of their sleep, which revealed that d*FMR1^B55^* generally sleeps more compared to the *w^1118^* and *per^01^
*phenotypes. Although there is a high frequency of rhythmic flies in the *dFMR1^B55^
*group in both sexes, it is important to emphasize that they mostly had a significantly lower power of rhythm than *w^1118^*. On the other hand, their power of rhythm is similar to rhythmic *per^01^
*flies. All investigated groups of flies had a similar period of CR, which was around 24 h. CR changes through the measurement of locomotor activity have been previously tested in a few *Drosophila* FXS models. All these studies showed *dFMR1* mutants as heterogeneous populations consisting of both rhythmic and arrhythmic individuals [24,26,28,29]. In accordance with our study, previous studies that investigated CR in *dFMR1^B55^* mutants also found differences in the frequency of rhythmicity between *dFMR1^B55^* and wild-type flies [24,29]. However, their results were independent of sexes, while groups of flies were divided by sexes in the current study. Although the frequency of rhythmicity of the *dFMR1^B55^
*mutants found in previous studies [24,29] was much lower than in the current study, these results cannot be compared due to different experimental designs. Specifically, in the current study, the experimental design was based on high-resolution video tracking of flies in subgroups divided by sex. However, similarly to our results, Inoue et al. (2002) calculated the period for the *dFMR1^B55^* mutants as 23.80 ± 0.50 h [24].

*Per* is a key gene in CR regulation [33] and *per^01^* was included as a control group in this study based on results in previously published articles that reported the majority of *dFMR1^B55^* males as arrhythmic [24]. Interestingly, according to the results obtained in the current study using high-resolution videography and VANESSA software, the majority of these flies were rhythmic with a low power of rhythm (see Figure 4). *per^01^
*mutants have a non-functional PER protein and cannot sustain rhythmicity properly [31]. However, there are studies where weak CR components were detected in *per^01^
*[34,35]. Klarsfeld et al. (2003) classified more than a third of *per^01^
*individuals as rhythmic [31], with weak power of rhythm, which is in concordance with the results obtained in the present study. In the current study, the frequency and power of *dFMR1^B55^
*males’ rhythmicity did not differ from *per^01^
*males, suggesting that regarding CR, *dFMR1^B55^* males are more similar to the *per^01^
*line, where weak rhythm was detected, and completely different than *w^1118^* of the same sex. Thus, the main research question is could scientists consider that *dFMR1^B55^* males are rhythmic with weak power of rhythmicity rather than they are being completely arhythmic? Based on our findings, it is evident that most *dFMR1^B55^* males exhibit weak rhythmicity compared to the control group *w^1118^*. This distinction in rhythmicity is supported by the methodology employed in our research. However, the same conclusion cannot be drawn for *dFMR1^B55^* females, as our results indicate that almost all of them displayed rhythmicity. Also, *w^1118^* females could not be a proper control group for comparison since, unexpectedly, the frequency of rhythmicity is lower in *w^1118^* females than in the *dFMR1^B55^
*of the same sex.

Other studies investigating CR in the *Drosophila* FXS model used different strains (*dFXR^Δ113^* and *dFMR^3^*) and also showed differences in the frequency of rhythmicity between FXS fly models and wild-type flies [26,28]. The power of rhythm as a CR parameter of *dFMR^3^* was only reported in the study conducted by Dockendorff et al. (2002) using Fast Fourier Transform (FFT) analysis. Similar to our results, they showed a much lower power of rhythmicity in *dFMR1* male mutants compared to wild-type [26]. In addition, previous sleep studies showed longer total sleep duration and a higher number of sleeping bouts in the *Drosophila* FXS model compared to the wild-type, which is consistent with our results [27]. Specifically, the current study was conducted using virgin male flies that were kept in isolation whereas females were kept in groups of five due to aggression in male individuals [36,37]. Although it is well known that isolation can influence and reduce sleep in fruit flies [38], our study revealed that both female and male *dFMR1* mutants generally slept more than control groups regardless of rearing conditions. In a previous study, no measures were made separately on females and males as was done in the present study. Our study shows that *dFMR1* male mutants spend more sleeping time than *dFMR1* female mutants. However, further study is needed to clarify differences between sexes in sleep patterns based on social rearing conditions.

All previously described CR studies of *Drosophila* FXS models used the infrared-based beam-crossing method by the *Drosophila* Activity Monitoring (DAM) system [24,26,27,28,29]. The previously used data analysis software was different in different studies. High-resolution continuous video monitoring with the Zantiks MWP Unit (https://zantiks.com/products/zantiks-mwp accessed on 8 July 2024) used in the current study is different from the most widely used infrared-based beam-crossing method. While the infrared-based method considers the number of crossing beam events during time and is limited to movements near the beams, the results of videography are presented as crossed distances. Micromovements of *Drosophila* happening anywhere in the well can also be detected using this tracking system. Some scientists reported their concerns about the under- or over-estimation of infrared detection of some fly movements. These limitations are linked to fly positions in the system [39]. In the current study, we can detect frequent short grooming bouts followed by short, crossed distances in *dFMR1^B55^
*mutants showing that the video monitoring used in this study can detect those micromovements anywhere in the system. These micromovements could be probably the main reason for a higher frequency of rhythmic flies in *per^01^* groups and generally less total sleep values. The observed difference in methodology between the DAM system and video-based monitoring may be also the main reason for variable results on CR in *dFMR1* mutants. Video monitoring as a more sensitive method could be useful, especially in mutants with reduced overall motility [33]. Bolduc et al. (2010) and Stojkovic et al. (2024) reported lower locomotor activity in *dFMR1^B55^* adults compared to *w^1118^* [25,40], which is similar to the current study and could be related to significantly higher total sleeping in *dFMR1^B55^
*mutants. Video recording always shows less sleep than DAM systems [41]. Micromovements less than 3 mm, i.e., activity records less than 1 body length of flies, will be considered for exclusion in the next iteration of our work. However, based on current and previous results, video monitoring used in our study could present a recommended methodology for sleep research and CR analyses in *dFMR1* mutants. It is important to emphasize that this study was not aimed at characterizing “more minute microbehavior” during sleep, as was done by Keleş et al. (2023) in Mark Wu’s laboratory [42]. Characterizing these “more minute microbehaviors” requires high-resolution and much closer monitoring of single flies and all their body parts; it is a novel approach but is currently limited by advancements in high-throughput versus high-zoom video acquisition [42]. However, our study emphasizes that video analysis could be more precise than traditional DAM systems. In brief, we aimed to develop an affordable replacement for the high-throughput DAM system with a similarly high-throughput, yet more sensitive, video recording system. Despite the study advantages, our research still primarily measured locomotor activity, but at a higher resolution, and differs somewhat from other video analyses like those done by Zimmerman et al. (2008), which conducted video analyses in a 1D environment (tubes) [43].

The Zantiks MWP Unit is a fully automated system with no need for additional interventions during the whole experimental period [44]. The system includes software, is controlled via a web browser, and has full customer support. On the other hand, challenges are associated with software for CR and sleep analyses because almost all available software is adapted to the DAM system. However, VANESSA is an open-source, user-friendly application for CR and sleep data acquisition, analysis, and visualization [45,46], and it was adapted for analyses of fruit fly video tracking recordings in our research. The VANESSA apps have Graphical User Interfaces (GUIs), are also hosted on a server, and can be directly used from a browser (available on https://cryptodice.shinyapps.io/vanessa-dam-cra/ (accessed on 8 July 2024) and https://cryptodice.shinyapps.io/vanessa-dam-sa/ (accessed on 8 July 2024)) [46]. Neither Zantiks MWP Unit nor VANESSA apps require high additional engineering or programming knowledge, which is an important advantage for research laboratories. The combination of Zantiks MWP and VANESSA represents a modern and useful technology that provides a more sensitive methodology in the field of chronobiology.

Finally, although the current study was conducted on the *Drosophila* FXS model, the biological significance of the findings, particularly their relationship with the clinical aspects of FXS, could be important. For example, due to differences in study design and instruments of assessment, sleep problems were reported in 27–77% of individuals with FXS [47,48,49,50,51,52]. These problems in children with FXS were described as mild to moderate [49]. The study conducted by Budimirovic et al. (2022) assessed the relationship between sleep difficulties and behavioral problems in children with FXS. As reported, almost all sleep problems were significantly associated with the presence of irritability/aggression, and hyperactivity, and some of them were associated with hypersensitivity [19]. In addition, sleep problems are associated with impaired circadian clock function in individuals with neurodevelopmental problems, including FXS [53,54]. Furthermore, sleep and CR impairments were also reported in the study using a mouse model of FXS. Specifically, Saré et al. (2017) described deficient sleep in *Fmr1* KO mice [55]. Interestingly, our findings of increased sleep time in the *Drosophila* FXS model are in contrast with results in the *Fmr1* KO mice, in which sleep duration was decreased. On the other hand, our findings on sleep behavior are in line with previously obtained results in fruit flies in studies with *Drosophila* FXS models [24,27]. These phenotypic variations might be due to the lack of both *Fmr1* paralogs, *Fxr1* and *Fxr2*, in flies [55]. Comprehensively, the results of research using different animal models of FXS confirmed that sleep problems are linked with CR impairments [56]. Although sleep problems and CR impairment were described in humans with FXS and different animal models, they are not very well characterized and the nature of sleep and CR abnormalities is not well understood. However, the current study on the *Drosophila* FXS model did not provide a direct assessment of fly behavior related to sleep changes and CR impairments but increased the pool of knowledge on sleep and CR in FXS. Thus, this work could make a significant contribution to the field. Additionally, given the absence of treatment data, this study should be viewed as a foundational step for future research focused on improving the identification and management of sleep issues in FXS.

There are some limitations of the current study that could be addressed in future research. For example, as explained in the Methods section, the rearing conditions differed for male and female flies due to aggression observed in males. In addition, micromovements of the flies that were less than 3 mm (approximately one body length) were not excluded from the analyses; these movements will be considered for exclusion in our future work. Finally, it will be interesting to associate sleep and CR changes with behaviors in *dFMR1^B55^* mutants. Overcoming these limitations will lead to more precise findings. Overall, the approach is novel and potentially offers more sensitive and detailed results compared to traditional methods.

## 4. Materials and Methods

### 4.1. Flies

The *dFMR1^B55^
*mutants were created by Inoue et al. in 2002 through an imprecise excision of the EP(3)3422 P-element. This process resulted in a deletion of the *dFMR1* genomic DNA, which included exons 2, 3, and 4, thereby creating a protein null allele B55 [24]. The control groups for this study consisted of the wild-type *white^1118^
*(*w^1118^)* and *period^01^* (per^01^) mutants. In order to justify the choice of control groups and explain how the study findings are being validated, it is important to emphasize that *w^1118^* is the host of the P-element that was excised to generate the *B55* allele [24,26,57]. *Per^01^* was used as a standard control group for CR and sleep pattern analysis.

Flies were reared under a 12 h light/12 h dark cycle on standard cornmeal/molasses/agar medium at 25°C and 60% relative humidity [58]. Virgin flies were collected 8 h after eclosion. Each virgin male was kept separately in its own vial due to their aggressive behavior [36,37] while virgin females were kept in groups of five on standard cornmeal/molasses/agar medium. After flies reached the age of 3–4 days, video monitoring of locomotor activity started at 7 a.m.

All analyzed groups of flies were divided based on phenotype and sex and each group contained 50 flies in total.

### 4.2. Video Monitoring of Locomotor Activity

Video monitoring and data collection of locomotor activity was performed in the Zantiks MWP Unit (Zantiks Ltd., Cambridge, UK). Demo scripts for Zantiks MWP Unit are available on the Zanscript-Script library Zantiks. A total of 96 flies (virgin males and females) were individually housed in each well of a 96-well plate. Each well was partly filled with a transparent sugar/agar medium. Animals were video tracked under standard conditions of 25 °C and 55–65% humidity. Locomotor activity rhythms were measured as distance traveled (mm) every 30 s under light–dark cycles of 12 h (LD12:12; light at 7.00 am marked as ZT00, and dark at 7.00 pm marked as ZT12) for 2 days followed by constant darkness (dark–dark, DD) for 8 days. In other words, the first two experimental days presented the light–dark (LD) period and consisted of daytime (DT) and nighttime (NT). In addition, the 3rd to 10th experimental days presented the dark–dark (DD) period.

The camera is built into the Zantiks system and is a 1440 × 1080 camera based on the Sony CMOS Pregius IMX273 sensor (software version: 2022-06-07, console version: 2022-05-07). The output is normally binned to 720 × 540, and the camera is run at 30 frames per second. Output could be at 1440 × 1080, but the frame rate would be reduced. Lighting is in InfraRed, and the exposure time is 3 ms, with a gain of 0. Flies are tracked every frame to give X and Y coordinates and the distance traveled can then be computed (subject to a minimum move tolerance) and output at any desired rate (e.g., distance traveled per second, per 30 s, or per 5 min). Tracking is based on a background subtraction algorithm, and is carried out in real time within the unit—no external computing resources are required. The video recording is from all 96 chambers simultaneously. Real-time video can be stored on the unit (100 + GB capacity), but normally we store a timelapse video with a trace of the track on it as a visual record (the whole experiment is around 500 MB).

### 4.3. CR and Sleep Pattern Analyses

Analyses of parameters were performed by VANESSA, an open-source R-based set of applications for CR and sleep data acquisition, analysis, and visualization (available on https://github.com/orijitghosh/VANESSADAM (accessed on 8 July 2024)). For CR analyses, frequency of rhythmicity, period (certain time in which cycles are repeated naturally), and power of rhythmicity (strength of CR) were analyzed.

Free running periods and powers of the CR were estimated by the chi-square periodograms in the highly customized version of VANESSA made to analyze data from Zantiks MWP. The fly was considered rhythmic if the peak of the periodogram appeared above the alpha = 0.05 confidence level. Males and females were analyzed separately.

For sleep analyses, sleep was defined as behavioral inactivity lasting ≥5 min, and the following parameters were analyzed in VANESSA, separately for DT and NT in the LD period, as well as values for the whole day in the DD period:Number of bouts: number of sleeping episodes;Sleep bout length: duration of bout (i.e., sleeping episode);Day sleep latency: the duration between ZT00 and first sleeping bout after ZT00;Night sleep latency: the duration between ZT12 and first sleeping bout after ZT12;Total sleep time (TST): time spent sleeping;Wake after sleep onset (WASO: the number of minutes a fly spends awake after having initially fallen asleep and is calculated using the formula: total night time from ZT12 to ZT24 (720 min) − (TST in NT–night sleep latency).Sleep fragmentation index (SFI) calculated using the formula: number of bouts/TST.

All sleep parameters are presented as value per day.

### 4.4. Statistical Analyses

Statistical analyses were performed in GraphPad Prism 8.0 (GraphPad Software, Inc., San Diego, CA, USA). Data are shown as both mean ± SD and median values with range. All statistical analyses were separately performed for different sexes. For normal distribution testing, Shapiro–Wilk and Kolmogorov–Smirnov tests were used. Comparison of the frequency of rhythmicity was performed using the chi-square test. For all other analyzed parameters, the Kruskal–Wallis test with Dunn’s multiple comparison (alpha = 0.05) was used. Specifically, the Mann–Whitney test was only used for differences between sexes of the same phenotype. Significance was defined as *p* ≤ 0.05.

## 5. Conclusions

CR and sleep analyses by new tools consisting of high-resolution videography and a highly customized version of open-source software provide more sensitive results in the analysis of *dFMR1* mutants, which could be crucial for all further research in this model of fruit flies. Specifically, this study revealed that *dFMR1* male mutants can be considered weak rhythmic flies rather than totally arrhythmic and represent a good experimental model for further behavioral research in the field of FXS. In addition, *dFMR1* male mutants could also represent a good model for further sleep research based on their sleep behavior. Finally, *dFMR1^B55^
*male mutants could be an important tool for pharmacological initial drug screening as an example of an alternative animal model of a genetic disorder, which includes impairment of CR and sleep behavior. The combination of affordable videography and software used in the current study is a significant improvement over previous methods and will enable broader adaptation of such high-resolution behavior monitoring methods.

## Figures and Tables

**Figure 1 ijms-25-07949-f001:**
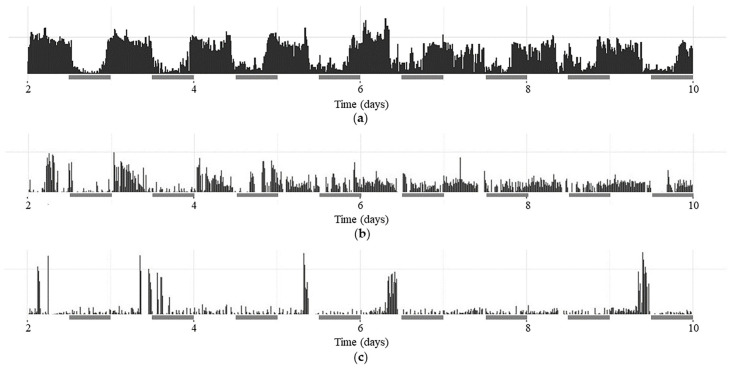
Presentation of individual actograms from the 3rd to the 10th experimental day during dark–dark (DD) period for (**a**) rhythmic *w^1118^* male; (**b**) arhythmic *per^01^
*male; (**c**) arhythmic *dFMR1^B55^* male. Actograms of females are not presented due to less evident differences among them. Gray bars indicate subjective days under DD.

**Figure 2 ijms-25-07949-f002:**
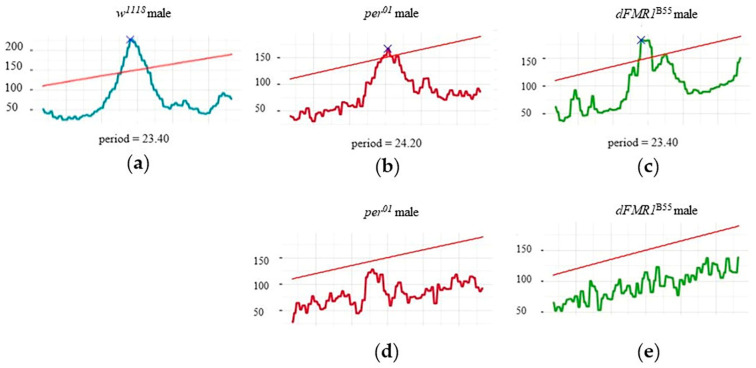
Presentation of representative individual chi-square periodograms for (**a**) rhythmic *w^1118^* male; (**b**) rhythmic *per^01^
*male; (**c**) rhythmic *dFMR1^B55^* male; (**d**) arhythmic *per^01^
*male and (**e**) arhythmic *dFMR1^B55^* male. The red oblique l line depicts the alpha = 0.05 threshold.

**Figure 3 ijms-25-07949-f003:**
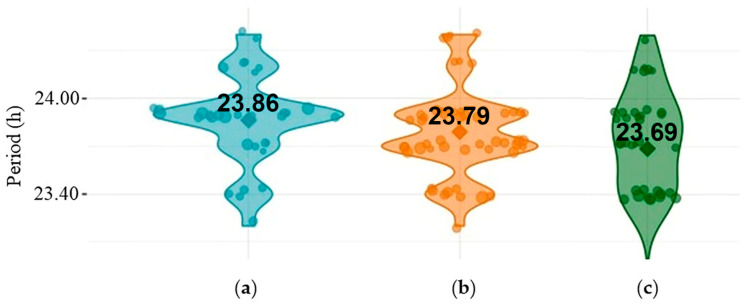
Violin plots showing the distribution of periods of (**a**) rhythmic *w^1118^* males; (**b**) rhythmic *per^01^
*males; (**c**) rhythmic *dFMR1^B55^* males.

**Figure 4 ijms-25-07949-f004:**
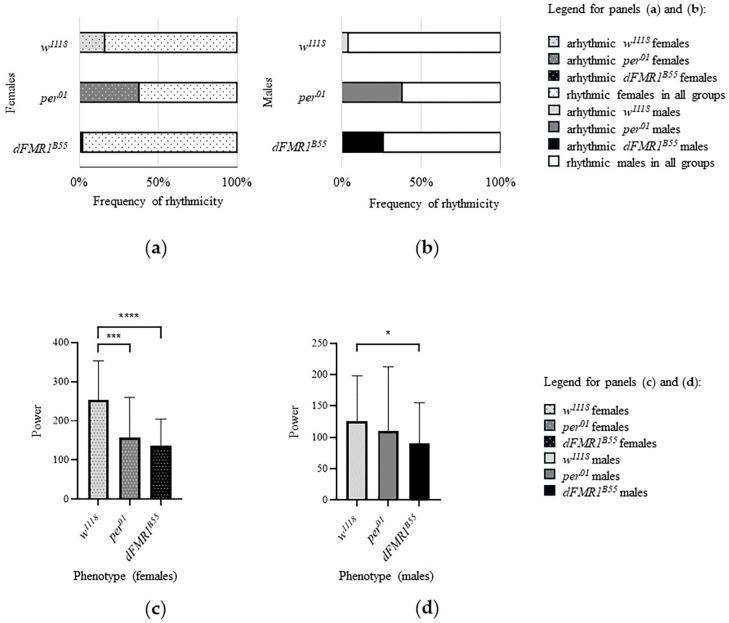
Bar plots of CR parameters in *w^1118^*, *per^01^*, and *dFMR1^B55^
*rhythmic flies. (**a**) frequency of rhythmicity in females; (**b**) frequency of rhythmicity in males; (**c**) power of rhythm in females; (**d**) power of rhythm in males. Error bars, as graphical representations of the variability of data, represent the standard deviations of data sets relative to the mean. Statistically significant differences are presented as: (*) *p* < 0.05, (***) *p* < 0.001 and (****) *p* < 0.0001.

**Figure 5 ijms-25-07949-f005:**
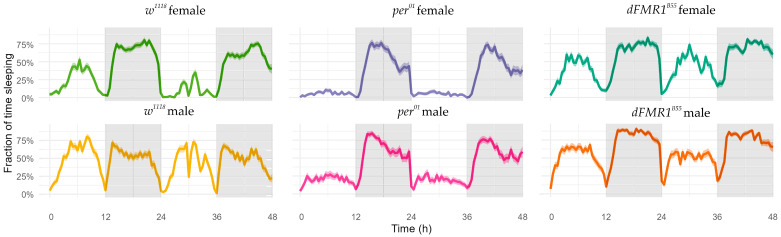
Average sleep profiles (fraction of time sleeping) for *w^1118^*, *per^01^*, and *dFMR1^B55^* flies (female and male) in light–dark (LD) period (1st to 2nd day).

**Table 1 ijms-25-07949-t001:** Sleep parameters for female and male flies in *w^1118^, per^01^*, and *dFMR1^B55^
*groups in daytime (DT; ZT00-ZT12; lights-on) part of the day in the light–dark (LD) period of experiments.

Sleep Parameter	Sex	Females				Males			
	Phenotype	*w^1118^*	*per^01^*	*dFMR1^B55^*	*p* Value	*w^1118^*	*per01*	*dFMR1^B55^*	*p* Value
No of bouts	Mean ± SD	10.18 ± 5.92	7.21 ± 7.55	16.15 ± 6.83		16.11 ± 5.63	12.79 ± 8.34	20.50 ± 8.68	
	Median [min–max]	10 [1–29]	5 [1–32]	16 [1–31]	<0.0001	16 [1–29]	11 [1–34]	22 [2–38]	<0.0001
Sleep bout length (min)	Mean ± SD	11.97 ± 6.51	7.45 ± 2.94	16.92 ± 6.63		22.97 ± 16.37	10.68 ± 6.09	18.05 ± 10.40	
	Median [min–max]	10.0 [5–36.33]	6.50 [5–18.60]	16.65 [5–36.18]	<0.0001	16.24 [6.63–95.60]	8.33 [5–44.24]	16.47 [5.17–60.78]	<0.0001
Sleep latency (min)	Mean ± SD	237.45 ± 139.95	223.09 ± 226.05	147.15 ± 88.87		127.64 ± 86.61	135,32 ± 104.08	89.10 ± 83.37	
	Median [min–max]	241 [0–666]	153 [1–1440]	135.50 [0–435]	<0.0001	127.50 [0–489]	100 [0–509]	67 [0–453]	<0.0001
TST (min)	Mean ± SD	119.25 ± 120.70	38.67 ± 70.64	270.44 ± 133.37		318.26 ± 155.96	138.73 ± 150.49	357.65 ± 169.94	
	Median [min–max]	81 [0–647]	8.50 [0–370]	300 [0–499]	<0.0001	325 [0–617]	82.50 [0–694]	414.50 [13–606]	<0.0001

Abbreviations: *w^1118^-white^1118^; per^01^-period^01^*; *dFMR1^B55^*-*dFMR1* mutant flies; min-minute; TST-total sleeping time; *p* value < or = 0.05 is considered statistically significant.

**Table 2 ijms-25-07949-t002:** Sleep parameters for female and male flies in *w^1118^, per^01^* and *dFMR1^B55^
*groups in nighttime (NT; ZT12-ZT24; lights-off) part of the day in the light–dark (LD) period of experiments.

Sleep Parameter	Sex	Females				Males			
	Phenotype	*w^1118^*	*per01*	*dFMR1^B55^*	*p* Value	*w^1118^*	*per01*	*dFMR1^B55^*	*p* Value
No of bouts	Mean ± SD	21.53 ± 6.91	10.18 ± 5.18	17.10 ± 8.41		23.30 ± 8.76	14.34 ± 4.95	15.96 ± 10.14	
	Median [min-max]	22 [6–37]	9.50 [2–30]	16 [2–43]	<0.0001	23.50 [1–44]	15 [5–32]	14 [1–47]	<0.0001
Sleep bout length (min)	Mean ± SD	22.37 ± 14.60	40.58 ± 27.35	37.73 ± 38.03		15.63 ± 8.52	32.15 ± 16.49	73.40 ± 109.02	
	Median [min-max]	17.19 [5.67–87.71]	34.03 [6.59–142]	27.42 [9.67–286.5]	<0.0001	12.66 [5–41]	29.28 [9.09–101.14]	40.75 [9.26–656]	<0.0001
Sleep latency (min)	Mean ± SD	94.20 ± 48.81	136.86 ± 51.86	109.84 ± 81.30		67.98 ± 68.28	100.99 ± 49.87	78.36 ± 46.62	
	Median [min-max]	94.50 [4–376]	134.50 [51–261]	110 [1–514]	<0.0001	54.50 [6–576]	101.5 [2–230]	86 [1–179]	<0.0001
TST (min)	Mean ± SD	409.81 ± 127.62	332.60 ± 146.64	447.26 ± 102.53		348.66 ± 162.56	405.44 ± 136.74	520.98 ± 81.72	
	Median [min-max]	416 [34–618]	324 [18–630]	458.50 [57–622]	<0.0001	347 [5–638]	418 [97–699]	537 [231–672]	<0.0001
WASO (min)	Mean ± SD	404.40 ± 161.20	524.3 ± 176	382.60 ± 152.20		439.30 ± 199.10	415.5 ± 174.3	277.40 ± 83.40	
	Median [min-max]	392.50 [112–1009]	541.50 [153–962]	351 [122–1042]	<0.0001	416 [93–1291]	408 [24–812]	272.50 [83–538]	<0.0001
SFI	Mean ± SD	0.06 ± 0.03	0.04 ± 0.03	0.04 ± 0.02		0.08 ± 0.04	0.04 ± 0.02	0.03 ± 0.03	
	Median [min-max]	0.06 [0.01–0.18]	0.03 [0.01–0.15]	0.04 [0–0.11]	<0.0001	0.08 [0.02–0.2]	0.04 [0.01–0.11]	0.02 [0–0.11]	<0.0001

Abbreviations: *w^1118^-white^1118^; per^01^-period^01^*; *dFMR1^B55^*-*dFMR1* mutant flies; min- minute; TST-total sleeping time; WASO-wake after sleep onset; calculated as total nighttime from ZT12 to ZT24 (720 min) − (TST in NT–night sleep latency); SFI-sleep fragmentation index calculated as number of bouts/TST; *p* value < or = 0.05 is considered statistically significant.

## Data Availability

The raw data supporting the conclusions of this article will be made available by the authors on request.

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
