# Peer review of "Circadian Rhythm and Sleep Analyses in a Fruit Fly Model of Fragile X Syndrome Using a Video-Based Automated Behavioral Research System"

_ijms, 2024, doi:10.3390/ijms25147949_

Round 1

Reviewer 1 Report

Comments and Suggestions for Authors

Feedback for the Manuscript: "Circadian rhythm and sleep analyses in a fruit fly model of Fragile X Syndrome using a video-based automated behavioral research system"

General Comments

This manuscript addresses an important area of research in Fragile X Syndrome (FXS) using an innovative video-based platform for analyzing circadian rhythm (CR) and sleep patterns in a Drosophila melanogaster model. The approach is novel and potentially offers more sensitive and detailed results compared to traditional methods.  However, some aspects need improvement in data analysis, biological interpretation, and methodological discussion. This work could make a significant contribution to the field. Addressing the points raised below will enhance the manuscript's clarity, impact, and scientific rigor, making it a valuable resource for researchers studying FXS and related neurodevelopmental disorders.

Major Points to Address

  1. The manuscript introduces a novel video-based platform for CR and sleep analysis in Drosophila, which is a significant advancement. However, the findings related to weak rhythmicity in dFMR1B55 mutants are not entirely novel, as similar observations have been reported previously. I suggest the authors emphasize how this new method improves upon existing technologies. In its current form, the manuscript briefly mentions the novelty of the video-based platform but does not delve deeply into its methodological advantages. Provide a more detailed comparison with traditional methods, highlighting specific advantages such as sensitivity, accuracy, or the ability to capture subtle behavioural changes that were previously undetectable and any new insights gained that were not possible with older methods like the Trikinetics system.
  1. The current analysis focuses on rhythmicity, period, and power of rhythm. However, additional sleep parameters could provide a more comprehensive understanding of sleep disturbances in the FXS model. I propose to analyze and report additional sleep parameters such as sleep bout length, sleep latency, wake after sleep onset (WASO), total sleep time (TST), and sleep fragmentation index, common in the field of Drosophila. This would give a more holistic view of the sleep patterns and their disruptions in the FXS model. However, I am unsure if this is possible with the Zantic syste.
  1. The manuscript lacks detailed descriptions of the statistical methods used for analyzing the data. Please provide a thorough description of the statistical analyses, including any corrections for multiple comparisons (e.g., Bonferroni, Holm). Justify the choice of control groups and explain how they validate the findings. Including more replicates or different control strains could strengthen the results.
  1. The biological significance of the findings, particularly how they relate to neurodevelopmental aspects of FXS, is not well-explained. The authors need to discuss the implications of the observed CR and sleep disruptions in the context of FXS pathology. How do these findings relate to the cognitive, social, motor and behavioural symptoms of FXS? Integrate comparisons with other FXS models (e.g., mouse models) to provide a broader perspective on the translational potential of the Drosophila model. How can the authors exclude the motor deficits in FXS flies as the main factor for the phenotypes they observed?

Minor Points to Address

1.     The manuscript contains minor grammatical errors and inconsistencies in terminology. Ensure consistent use of scientific terminology (e.g., italicize "Drosophila melanogaster").

2.     The resolution in some images is really bad and needs to be improved.

3.     Some statements, particularly those related to the prevalence and impact of FXS, lack appropriate citations. Update the reference list to include recent and relevant studies, especially those discussing the use of Drosophila models in FXS research.

Comments on the Quality of English Language

  • The manuscript contains minor grammatical errors and inconsistencies in terminology.
  •  

Author Response

Major Points to Address

1. The manuscript introduces a novel video-based platform for CR and sleep analysis in Drosophila, which is a significant advancement. However, the findings related to weak rhythmicity in dFMR1B55 mutants are not entirely novel, as similar observations have been reported previously. I suggest the authors emphasize how this new method improves upon existing technologies. In its current form, the manuscript briefly mentions the novelty of the video-based platform but does not delve deeply into its methodological advantages. Provide a more detailed comparison with traditional methods, highlighting specific advantages such as sensitivity, accuracy, or the ability to capture subtle behavioural changes that were previously undetectable and any new insights gained that were not possible with older methods like the Trikinetics system.

Thank you for your observation. Please note that added more methodological details (please see lines: 505 -517). Also, we discussed detailed comparison with traditional methods (please see lines: 411 -420). In addition, we mentioned that that this study was not aimed at characterizing 'more minute microbehavior' during sleep, as was done by Keleş et al. (2023) in Mark Wu's laboratory [Ref #42]. Characterizing these 'more minute microbehaviors' requires high-resolution and much closer monitoring of single flies and all their body parts; it is a novel approach but currently limited by advancements in high-throughput versus high-zoom video acquisition (please see lines: 409-414).

Finally, we added study limitation (please see lines: 465 -473), and changed the last sentence in Abstract (please see lines: 37 - 39) and in Conclusion (please see lines: 563 – 565).

2. The current analysis focuses on rhythmicity, period, and power of rhythm. However, additional sleep parameters could provide a more comprehensive understanding of sleep disturbances in the FXS model. I propose to analyze and report additional sleep parameters such as sleep bout length, sleep latency, wake after sleep onset (WASO), total sleep time (TST), and sleep fragmentation index, common in the field of Drosophila. This would give a more holistic view of the sleep patterns and their disruptions in the FXS model. However, I am unsure if this is possible with the Zantic system.

Thank you for your suggestion. We added more comprehensive and very detailed sleep analyses. Please see: Methods (lines 527 - 540), Results (lines 206 - 318), new Tables 1 and 2, Supplementary Table 1and new Figure 5. Accordingly, the Discussion section is improved (see whole Discussion, lines 320 – 473).

3. The manuscript lacks detailed descriptions of the statistical methods used for analyzing the data. Please provide a thorough description of the statistical analyses, including any corrections for multiple comparisons (e.g., Bonferroni, Holm). Justify the choice of control groups and explain how they validate the findings. Including more replicates or different control strains could strengthen the results.

Thank you for your suggestion. We provided description of the statistical analyses. Please see lines: 541 -551. In addition, justification of the choice of control groups is added in the current version of the manuscript. Please see lines: 480 -483.

 4. The biological significance of the findings, particularly how they relate to neurodevelopmental aspects of FXS, is not well-explained. The authors need to discuss the implications of the observed CR and sleep disruptions in the context of FXS pathology. How do these findings relate to the cognitive, social, motor and behavioural symptoms of FXS? Integrate comparisons with other FXS models (e.g., mouse models) to provide a broader perspective on the translational potential of the Drosophila model. How can the authors exclude the motor deficits in FXS flies as the main factor for the phenotypes they observed?

Thank you for your observation. As you suggested, we added comprehensive discussion on the biological significance of the findings, particularly how they related to neurodevelopmental aspects of FXS. We also provided integrated comparisons with other FXS models (e.g., mouse models). Please see Discussion, lines: 437 – 464.  

Minor Points to Address

  1. The manuscript contains minor grammatical errors and inconsistencies in terminology. Ensure consistent use of scientific terminology (e.g., italicize "Drosophila melanogaster").

Thank you for your observation. We corrected minor grammatical errors and inconsistencies in terminology. Please find our changes in whole manuscript.

  1. The resolution in some images is really bad and needs to be improved.

The resolution of all images is improved in this version of the manuscript (600 dpi for all figures). Please note that old Fig. 5 is excluded, and a new Fig. 5 is included.

  1. Some statements, particularly those related to the prevalence and impact of FXS, lack appropriate citations. Update the reference list to include recent and relevant studies, especially those discussing the use of Drosophila models in FXS research.

Thank you for your comment. Please note that new references are added, and the part on FXS is improved (please see text between lines 46 -87).

Reviewer 2 Report

Comments and Suggestions for Authors

The manuscript of Dr. Sara Milojevic, Dr. Dragana Protic, and coauthors is a methodological paper promoting a novel approach for evaluating circadian rhythms and sleep patterns in drosophila. Here, the drosophila is used as an animal model of Fragile X-syndrome. The main result is that dFMR1 male mutants can be considered weak rhythmic flies, unlike previous description in the literature. The demonstration is done with software and behavioral experiments without standard methods in molecular biology.

From my understanding, the paper is out of the scope of International Journal of Molecular Science. The authors have to be redirected to another MDPI journals more in line with chronobiology or software instrumentation.

I made a note to the editors to underline the problem and leave open the possibility to accept the paper with minor modifications if considered within the scope of IJMS.

The English style is fine.

Comments on the Quality of English Language

The English style is fine.

Author Response

Comment: The manuscript of Dr. Sara Milojevic, Dr. Dragana Protic, and coauthors is a methodological paper promoting a novel approach for evaluating circadian rhythms and sleep patterns in drosophila. Here, the drosophila is used as an animal model of Fragile X-syndrome. The main result is that dFMR1 male mutants can be considered weak rhythmic flies, unlike previous description in the literature. The demonstration is done with software and behavioral experiments without standard methods in molecular biology.

From my understanding, the paper is out of the scope of International Journal of Molecular Science. The authors have to be redirected to another MDPI journals more in line with chronobiology or software instrumentation.

I made a note to the editors to underline the problem and leave open the possibility to accept the paper with minor modifications if considered within the scope of IJMS.

The English style is fine.

Response:

Dear Reviewer,

 Thank you for your comment. However, this manuscript was transferred to Special issue "Current Molecular Science of Fragile X Syndrome and Associated Disorders" edited by Professor Flora Tassone and Professor Randi Hagerman”. As FXS is in our focus, we hope that current version of the manuscript is within the scope of this special issue of IJMS.

Reviewer 3 Report

Comments and Suggestions for Authors

I have enclosed my comments. I request authors to kindly address my concerns so that I can recommend this article for publication.

Is it possible to show the quantification for daytime and nighttime sleep? It is likely that both daytime and nighttime sleep has been increased in dFMR1B55 flies. Also, authors can calculate average sleep bout length during the day and nighttime to show that sleep has been increased significantly in dFMT1B55 flies. I request authors to add some sleep parameters, such as bout length, number of bouts, etc.

It seems that the rearing conditions differed for male and female flies. Male flies were kept in isolation whereas females were kept in groups of five. If that is true, why were two different social contexts used for this study? We know that social isolation can influence sleep. I request author to clarify this in the discussion. Also, please mention whether the female flies were mated when sleep measurements were carried out, as reproductive status can influence sleep.

I request authors provide details on the video analysis: specification of the camera used for fly tracking, the acquisition parameters, etc. Also, it would be nice to discuss the significance of the video analysis. Were the authors aiming to characterize micro behaviors associated with sleep as mentioned in the recent BioRxiv manuscript from Mark Wu’s lab? How was video tracking more advantageous than using a conventional DAM syste? Please provide some discussion.

I request the authors provide details on the main challenges involved in video tracking. How much memory space is required for storing the video? Are you really storing the video or only collecting the activity data? A direct comparison between DAM data and data from the video tracking system is missing. It would be nice to have this in the supplement.

It is a bit hard to follow the descriptions and figures. Especially when rhythmicity has been discussed for Per mutants, I would expect that the corresponding figure should be mentioned alongside. This will help readers to follow the logic. Also, I request authors to mention how sleep has been defined. In flies, most studies have relied on a definition of sleep as behavioral inactivity lasting >= 5 min. Please mention what definition and parameters were followed.

Author Response

  1. Is it possible to show the quantification for daytime and nighttime sleep? It is likely that both daytime and nighttime sleep has been increased in dFMR1B55 flies. Also, authors can calculate average sleep bout length during the day and nighttime to show that sleep has been increased significantly in dFMT1B55 flies. I request authors to add some sleep parameters, such as bout length, number of bouts, etc.

Thank you for your suggestion. Please note that we added more comprehensive and very detailed sleep analyses. Specifically, according to your comments, we added separately analyses for daytime and nighttime. All requested sleep parameters are added.

Please see: Methods (lines 527 - 540), Results (lines 206 - 318), new Tables 1 and 2, Supplementary Table 1and new Figure 5. Accordingly, the Discussion section is improved (see whole Discussion, lines 320 – 473).

  1. It seems that the rearing conditions differed for male and female flies. Male flies were kept in isolation whereas females were kept in groups of five. If that is true, why were two different social contexts used for this study? We know that social isolation can influence sleep. I request author to clarify this in the discussion. Also, please mention whether the female flies were mated when sleep measurements were carried out, as reproductive status can influence sleep.

Thank you for your comment. We discussed the rearing conditions in males and females as well as social isolation. Please see Discussion, lines 373 – 382. We also clarify it in Methods (lines: 486 -487).

In addition, we also clarify that male and female were virgin. Please see lines: 484 – 489.

I request authors provide details on the video analysis: specification of the camera used for fly tracking, the acquisition parameters, etc. Also, it would be nice to discuss the significance of the video analysis. Were the authors aiming to characterize micro behaviors associated with sleep as mentioned in the recent BioRxiv manuscript from Mark Wu’s lab? How was video tracking more advantageous than using a conventional DAM syste? Please provide some discussion.

Thank you for your comments. We provided details on the video analysis: specification of the camera used for fly tracking, the acquisition parameters, etc. Please see lines: 505 -516.

In addition, significance of video-based analyses is discussed in lines 414 -420. Also, we mentioned that this study was not aimed at characterizing 'more minute microbehavior' during sleep, as was done by Keleş et al. (2023) in Mark Wu's laboratory. Please see lines: 411 - 414.

I request the authors provide details on the main challenges involved in video tracking. How much memory space is required for storing the video? Are you really storing the video or only collecting the activity data? A direct comparison between DAM data and data from the video tracking system is missing. It would be nice to have this in the supplement.

Thank you for your comments and observation. Please note that we provided details regarding video tracking (including memory space, videos storges, etc). Please find lines: 505 - 516.

It is a bit hard to follow the descriptions and figures. Especially when rhythmicity has been discussed for Per mutants, I would expect that the corresponding figure should be mentioned alongside. This will help readers to follow the logic. Also, I request authors to mention how sleep has been defined. In flies, most studies have relied on a definition of sleep as behavioral inactivity lasting >= 5 min. Please mention what definition and parameters were followed.

Thank you for your observation. Please note that all figures are improved, old Fig. 5 is excluded and there is a new Fig. 5 in this version of the manuscript. Also, some figures are mentioned in Discussion (i. e. Fig 4 in line 348). We hope that this version could be easier to follow the logic.

In addition, please note that we mentioned how sleep has been defined. Please see lines: 527 -528.

Round 2

Reviewer 2 Report

Comments and Suggestions for Authors

The manuscript of Dr. Sara Milojevic, Dr. Dragana Protic, and coauthors is in line with the scope of the Special issue, along the item "Phenotype research as it relates to the molecular basis of X syndrome". I can recommend the work for publication. However, a second opinion from a specialist of behavioral research would be of interest.

Reviewer 3 Report

Comments and Suggestions for Authors

The authors modified the manuscript as per my suggestions. I am happy to recommend this for publication.